

# Analysis of Flow in Complex Terrain Using Multi-Doppler Lidar Retrievals

Tyler Bell[1,2], Petra Klein[1,2], Norman Wildmann[3], and Robert Menke[4]

[1]School of Meteorology, University of Oklahoma, Norman, OK, USA
[2]Cooperative Institute for Mesoscale Meteorological Studies, University of Oklahoma, Norman, OK, USA
[3]Deutsches Zentrum für Luft- und Raumfahrt e.V., Münchener Str. 20, Oberpfaffenhofen, Germany
[4]Technical University of Denmark - DTU Wind Energy, Fredriksborgvej 399, Building 118, 4000 Roskilde, Denmark

**Correspondence:** Tyler Bell (tyler.bell@ou.edu)

**Abstract.** The Perdigão Field Experiment set out to study atmospheric flows in complex terrain and to collect a high-quality dataset for the validation of meso- and micro-scale models. An Intensive Observation Period (IOP) was conducted from May 1, 2017 through June 15, 2017 where a multitude of meteorological instruments were deployed in a study area with the unique feature of two nearly parallel, 5 km long ridges separated by a 1.4 km wide valley perpendicular to the prevalent wind directions in the region. An essential part of the instrumentation were scanning Doppler lidars (DL) strategically placed to capture flow features above the ridges and in the valley. The arrangement of DLs presented an opportunity to create virtual towers where range height indicator (RHI) scans of individual instruments intersected. By combining DLs it is possible to retrieve multiple snapshots of the wind field in the form of a virtual tower. In total, four virtual towers distributed along the valley are retrieved every 15 minutes. The virtual towers typically cover heights from 50 m to 600 m above the valley floor. The Perdigão project also included a network of meteorological towers of different heights with wind measurements at an exceptionally high density (55 towers with 195 sonic anemometers) that are critical for studying turbulent exchange processes but provide only limited information about the complex interactions between the flow inside the valley and higher up across the ridges. The virtual towers extend the range of traditional in-situ observations and can fill in low altitude areas where traditional lidar processing techniques may have trouble retrieving accurate wind speeds due to the high spatial flow variability and prevalence of significant vertical motions in complex terrain. Along with the wind speed and direction, uncertainties of the virtual tower retrieval were analyzed. A case study of a nighttime stable boundary-layer flow with wave features in the valley is presented to illustrate the usefulness of the virtual towers in analyzing the spatially complex flow over the ridges during the Perdigão campaign.

## 1 Introduction

Scanning Doppler lidar (DL) systems have proven to be useful in many different sectors of atmospheric study. They have been used for boundary layer meteorology (Klein et al., 2015; Fernando et al., 2015), wind energy research (Banta et al., 2015; Newman et al., 2016; Choukulkar et al., 2017), and other various fields of study (Sathe and Mann, 2013; Bonin et al., 2017). DLs measure the radial velocity along a beam in a high spatial and temporal resolution. Different scanning strategies



can give different insights into the flow field surrounding the DLs. For example, a plan position indicator (PPI) scan gives a representation of the spatial variability in the horizontal by scanning at a fixed elevation and only moving in azimuth, while a range height indicator (RHI) scan essentially gives a cross section of the flow by staying at a fixed azimuth, but changing elevation.

By applying assumptions to the flow, one can use these different scan strategies to derive the two-dimensional (2D) and three-dimensional (3D) wind from a single DL. The simplest techniques to derive the wind speed and direction are the velocity azimuth display (VAD) technique (Browning and Wexler, 1968) and the Doppler beam swinging (DBS) technique (Strauch et al., 1984). However, these techniques make the assumption that the wind is horizontally homogeneous in order to retrieve the wind speed and direction; this is often not the case in boundary layer meteorology and can introduce errors

into the wind estimates. One area where the assumption of horizontal homogeneity is likely invalid is in the study of complex terrain, where atmospheric flows can have large spatial heterogeneity (Bingöl et al., 2009; Bradley et al., 2015). Correction techniques such as Leosphere's Flow Complexity Recognition (FCR) algorithm try to correct errors introduced by terrain by using simple flow models (see https://www.ieawindtask32.org/wp-content/uploads/2017/11/GM2017-Leosphere-Windcube-FCR-measurements.pdf). Other techniques using a single DL have been developed to limit this assumption (Waldteufel and

Corbin, 1979; Wang et al., 2015).

While retrieving the 3D wind field from a single DL requires some potentially invalid assumptions, this can be solved by using multiple DLs with beams that intersect in the same volume of air. By combining multiple different radial wind vectors, one can solve the 3D transformation matrix to get the 3D wind vector without applying any assumptions to the flow. There are multiple ways that this can be done. For example, co-planar RHI scans were used in the Terrain-induced Rotor Experiment to

study rotors caused by mountains (Hill et al., 2010), multi-Doppler scan strategies were used in the Perdigão 2015 and 2017 experiments to measure wind turbine wake deficits in highly complex terrain (Barthelmie et al., 2018; Menke et al., 2018; Wildmann et al., 2018a, b), and virtual towers were used in the Joint Urban Experiment in 2003 (Calhoun et al., 2006).

Multi-Doppler measurements can augment more traditional observation strategies and are highly adaptable to an experiment's science objectives. Though some precision is lost due to volumetric averaging, the results are still quite accurate

(Damian et al., 2014; Debnath et al., 2017). However, higher levels of uncertainty have been found with more complicated scan strategies (Choukulkar et al., 2017). For multi-Doppler retrievals, the magnitude of uncertainty was directly related to the ability to precisely coordinate scans. Additionally, higher levels of uncertainty are present during unstable conditions (Newman et al., 2016).

This study looks to answer how well 2D and 3D multi-Doppler measurements perform in a complex setting and provide an

analysis of uncertainties that the terrain introduces to both multi-Doppler measurements and more traditional single-Doppler wind estimation techniques.



**Table 1.** Characteristics of the RHI scans performed by each DL used in this study

| Lidar ID | Lidar Number | Azimuth | Min. Elevation | Max Elevation | Update Time |
|----------|-------------|---------|----------------|---------------|-------------|
| OU CLAMPS | 131 | 318 ° | 7.5 ° | 175 ° | 15 min |
| DLR #1 | 172 | 236.1 ° | -7.4 ° | 45.1 ° | ~30 s |
| DLR #2 | 109 | 236.4 ° | 8.5 ° | 122 ° | ~30 s |
| WS2 | 102 | 54.7 ° | -18.8 ° | 15.7 ° | ~30 s |
| WS5 | 105 | 52.3 ° | -12.8 ° | 21.7 ° | ~30 s |
| WS6 | 106 | 42.2 ° | -16.8 ° | 17.7 ° | ~30 s |

## 2 The Experiment

The data presented in this study were collected during the Perdigão Field Campaign during the Spring and early summer of 2017, which is one of multiple experiments conducted in order to build the New European Wind Atlas (NEWA) (Mann et al., 2017). As wind energy becomes more popular, the need for models to accurately predict energy output from wind resources becomes increasingly important. The goal of NEWA is to provide a standard dataset of wind energy resources for Europe through the creation of new and improved meso- and micro-scale models and modeling techniques. One important aspect of developing these modeling techniques is verifying their accuracy against a high quality dataset. Therefore, one of the main goals in the creation of the NEWA is developing new modeling methodologies which have been tested against observations from various measurement campaigns conducted in different landscapes and climates.

To validate numerical models, detailed measurements of the flow at multiple scales are required. To this day, data collected from the Askervein Hill project in 1982 (Taylor and Teunissen, 1987) are still a standard dataset to validate models and to test how they handle flow over complex terrain. Some of the experiments taking place under the NEWA are meant to augment the measurements from Askervein while adding another layer of complexity with observation campaigns focusing on flow phenomena that current modeling techniques are known to have difficulties with (e.g. vegetation canopies, steep slopes, double ridge configuration).

The Perdigão Experiment is one of the measurement campaigns that took place for NEWA (Mann et al., 2017). Perdigão is a small municipality located in central Portugal that sits adjacent to two parallel ridges of nearly equal height with steep slopes separated by a valley called Vale de Cobrão. The ridge axes extend northwest to southeast and are approximately 1.4 km apart (Figure 1). Previous wind-tunnel and numerical studies (e.g. Lee et al., 1987; Grubišić and Stiperski, 2009; Rapp and Manhart, 2011) have often focused on the flow over sinusoidal hills. Two parallel ridges is the best approximation in nature to study flow over periodic, sinusoidal type terrain.

Additionally, there is a single 2 MW wind turbine located on the southwest ridge. Wind resources are significantly affected by terrain and land cover. For example, winds at hub height are much higher at the top of a hill than they are on the lee side of



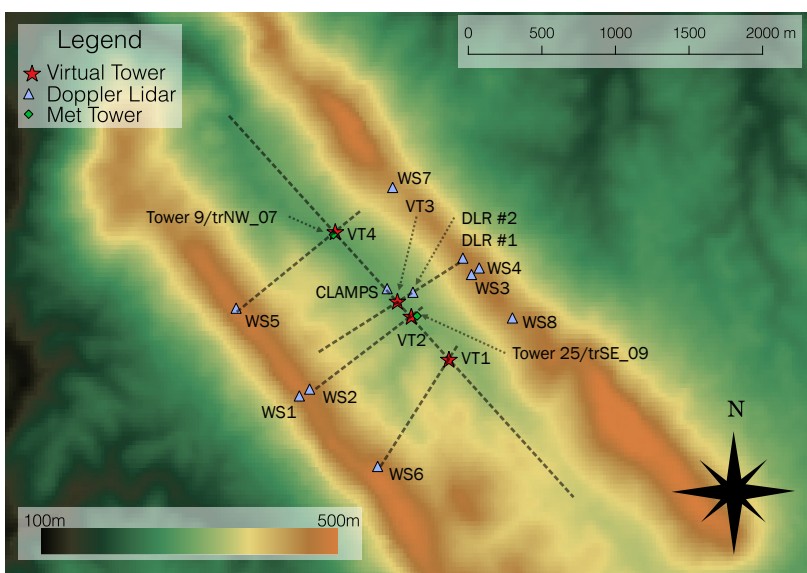

**Figure 1.** Map of DLs, meteorological towers, and virtual towers at the Perdigão site. The color fill corresponds to the height of the terrain above sea-level, red stars are virtual tower locations, blue triangles are DL locations, and green diamonds are instrumented meteorological towers. Tower 9/trNW_07 is a 60 m tower and Tower 25/trSE_09 is a 100 m tower.

the hill. The flow on the lee side of the hill, depends largely on the stability conditions, wind speeds, etc. These conditions are important when determining sites where wind turbines will consistently be able to produce energy.

Leading up to the Intense Observation Period (IOP) in the Spring of 2017, a meteorological tower had been operating on the SW ridge for a few years. This was used to construct a climatology of the wind directions over the ridge. According to the climatology, winds often were found to be directed perpendicular to the two ridges (Vasiljević et al., 2017). This provided a good opportunity to study the flow in a double-ridge setting. To gather the desired observations, a multi-national group of scientists from Europe and the United States converged in Perdigão to measure complex flow at unprecedented spatio-temporal scales. A combination of meteorological towers, DLs, radiometers, and other remote and in-situ platforms were dispersed throughout the valley (see Fernando et al., 2018).

## 2.1 Virtual Towers

For this study, multiple different DL configurations were jointly analyzed to retrieve the 3D wind vector in the valley. In total, 6 DLs from 3 different institutions were combined to retrieve virtual towers multiple times per hour. Figure 1 shows the location of the DLs considered in this study. Details about the scanning strategies for each DL can be found in Table 1.

The University of Oklahoma (OU) deployed the Collaborative Lower Atmospheric Mobile Profiling System (CLAMPS Wagner et al., in press) at the lower Orange site located inside the valley during the IOP. CLAMPS includes a Halo Photonics





Scanning DL that performed both cross- (NE to SW) and along-valley (NW to SE) RHI scans every 15 minutes. In addition, a 70 degree PPI scan was performed every 15 minutes preceding the RHIs. The remainder of the time, the DL was in stare mode to get vertical velocity statistics. CLAMPS also utilizes an Atmospheric Emitted Radiance Interferometer (AERI, Knuteson et al., 2004a, b) and a HATPRO Microwave Radiometer (Rose et al., 2005) for boundary layer temperature and humidity

profiling.

The University of Colorado (CU) operated a Leosphere V1 Windcube Profiling DL at the Lower Orange Site, very close the OU DL. The CU DL measures the wind speed and direction using the DBS technique (see Rhodes and Lundquist, 2013).

The German Aerospace Center (DLR) contributed three Leosphere Windcube 200S scanning DLs upgraded with the Technical University of Denmark's (DTU) WindScanner software. Two of these DLs performed continuous RHI scans in the cross-

valley direction, which resulted in an RHI approximately every 30 seconds. One of these DLs was located up on top of the NE ridge (DLR #1) and performed RHI scans to the SW, capturing one horizontal component of the wind. The other DL (DLR #2) was located on the slope of the NE ridge and performed RHIs to the SW as well.

In addition to the DLR DLs, DTU operated eight DLs of the same kind on top of the ridges. Six of these were configured to do co-planar scans inside the valley so the horizontal wind in the plane and the vertical velocity could be retrieved. These DLs

also operated in a continuous scan mode and produced a new RHI every 30 seconds.

The RHIs from DLR #1 and DLR #2 overlapped in a co-planar fashion, so by combining these DLs with the OU DL scans, it is possible to retrieve the three-dimensional wind field in the form of a virtual tower where the three planes intersect. Due to its positioning, DLR #2 was able to capture more of the vertical component of the wind in the location of the virtual tower, which allowed the retrieval of the three-dimensional wind vector.

Regarding the DTU DLs, only the DLs from each co-planar cross section that reached deeper into the valley were used. This resulted in 3 possible 2D horizontal wind retrievals using WS2, WS5, and WS6. The 2D retrievals assume there is no vertical velocity, and thus only provide the along- and cross-valley wind components.

In total, four virtual towers distributed along the valley are retrieved every 15 minutes when the CLAMPS DL performed its along-valley RHI. The virtual towers typically cover heights from 50m to 600m above the valley floor depending on the

minimum and maximum height of RHI intersection. None of the DLs used in the retrievals scanned in a coordinated fashion. However, overlapping sampling volumes provide an opportunity for testing the quality of virtual tower retrievals in complex flows.

### 3   Methods

Data from each DL were first converted to a common coordinate system. Once the XY location of the RHI intersection (i.e.

the virtual tower) was determined, minimum and maximum heights were manually determined for each of the virtual towers. The vertical spacing of the virtual tower points is 10 m. This spacing was selected to insure that each tower point had unique DL range gates to derive the 3D/2D winds. Using the location of these points, the azimuth ($\hat{\theta}_i$) and elevation ($\hat{\phi}_i$), relative to





each DL, was calculated and used to linearly interpolate the radial velocity from each DL to a new radial velocity at the point of the tower ($\hat{V}_{ri}$).

$$
\begin{bmatrix} u \\ v \\ w \end{bmatrix} = \begin{bmatrix} \sin\theta_1\cos\phi_1 & \cos\theta_1\cos\phi_1 & \sin\phi_1 \\ \sin\theta_2\cos\phi_2 & \cos\theta_2\cos\phi_2 & \sin\phi_2 \\ \sin\theta_3\cos\phi_3 & \cos\theta_3\cos\phi_3 & \sin\phi_3 \end{bmatrix}^{-1} \begin{bmatrix} \hat{V}_{r1} \\ \hat{V}_{r2} \\ \hat{V}_{r3} \end{bmatrix} \tag{1}
$$

where $u$ is the velocity in the east-west direction, $v$ is the velocity in the north-south direction, $w$ is the vertical velocity, $V_{ri}$
is the interpolated radial velocity from the DL, $\phi_i$ is the elevation, and $\theta_i$ is the azimuth angle.

Temporal resolution of the virtual towers is limited to the time resolution of the CLAMPS RHI. Since the scans were not coordinated to sample the same volume of space simultaneously, a time window needed to be determined. Choukulkar et al. (2017) used a time window of 15 s when comparing uncoordinated multi-Doppler retrievals to a sonic anemometer and found reasonable agreement. For this study, all the time periods considered take place overnight, which generally had more steady flow due to the lack of turbulent mixing. Therefore, a time window of 60 s was used. While most of the data fell well within this window, there were time periods late in the IOP where retrievals were not possible due to the CLAMPS RHI starting too late to be captured in the 60 s window.

## 3.1 Uncertainty Analysis

Possible uncertainties contained in the retrieval were analyzed using an idealized scheme derived from the methods discussed in Hill et al. (2010):

$$
\epsilon_{u_i} = \left[ \sum_{j=1}^{N} \left( \frac{\partial u_i}{\partial r_j} \right)^2 (\epsilon_{r_i})^2 \right]^{1/2} ; i = 1, 2, 3 \tag{2}
$$

where $\epsilon_{r_i}$ is the possible random error associated with each DL line-of-sight wind speed measurement, $u_i$ is the Cartesian velocity component of the wind, and $r_j$ is the radial velocity vector for each DL. This results in a formula for the error for each velocity component given a combination of the various azimuths, elevations, and ranges for each lidar. For this study, it was assumed that $\epsilon_{r_i}$ was .2 ms$^{-1}$ based on prior experience with the systems used. Results are shown in Figure 2. This analysis was done in a coordinate system that was aligned with the valley.

Note that with increasing height (i.e. as beams are scanning at a more vertical position), uncertainties in the horizontal wind components get larger and uncertainties in the vertical velocity get smaller. Also note that the errors in horizontal wind components in the cross-valley direction are much larger in the 3D retrieval. This is due to the proximity of VT3 to the OU DL; gates at much higher elevation angles must be used, meaning less of the horizontal component of the wind in the along valley direction is contained in the radial velocity.





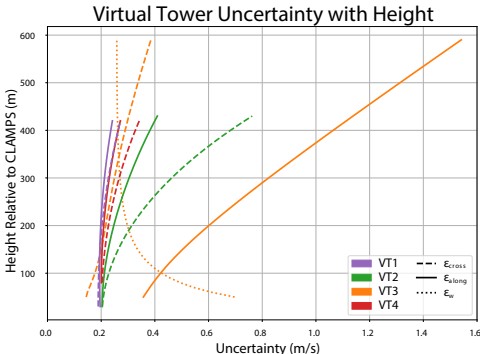

**Figure 2.** Uncertainty from the various virtual towers. Solid lines are uncertainty in along-valley direction, dashed lines are uncertainty in the cross-valley direction, and dotted lines are uncertainty in w. The color corresponds to the virtual tower (see Figure 1). Note that most of the uncertainty is contained in the along valley direction since the CLAMPS DL was scanning at a high elevation angle (elevations used ranged from $21°$ to $77°$). Additionally, the further away the virtual tower was from CLAMPS (i.e., the lower the elevation angle had to be), the lower the uncertainty.

## 3.2 Impact of Terrain on 2D towers

Since multiple virtual towers were only able to retrieve the horizontal wind vectors, it is important to quantify the amount of error this may cause given they were in complex terrain where large vertical velocities were often present. To do this, a theoretical idealized setup was constructed (Figure 3). DL1 and DL2 are fixed in place, but DL3 is allowed to move in range away from the tower, thereby decreasing the elevation angle required to observe the same point. This setup is meant to mimic the DL placements for the 3D tower (see VT3 in Figure 1). For 2D uncertainty calculations, only DL1 and DL3 were used.

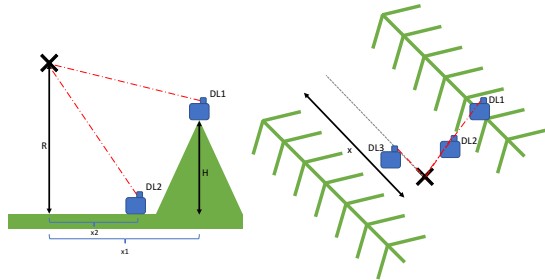

**Figure 3.** Idealized setup to examine the errors associated with neglecting the vertical velocity and only producing 2D virtual towers. The setup is meant to closely mimic the positions of the OU DL, DLR #1, and DLR #2 used in the 3D virtual tower. DL3 is varied along x to simulate different elevation angles.


One can then set a 3D wind speed and direction at the intersection of all the DL beams and calculate the radial velocities that are observed by each DL by rearranging Equation 1. By only using the radial velocities from DL3 and DL1, a 2D retrieval similar to the 2D retrievals done for the real virtual towers can be simulated (with one DL on top of the ridge and with the CLAMPS system inside the valley). However, these radial velocities will contain some component of vertical motion that will violate the assumptions used for the 2D towers therefor causing errors. In order to study the effect of the strong vertical velocities observed by the vertical stare from the CLAMPS DL, multiple vertical velocities were used to mimic the range of vertical velocities observed by the vertical stare from CLAMPS throughout the IOP.

Figure 4 shows expected errors in horizontal wind speed and direction that arise if the vertical component of the wind is assumed to be zero. It becomes immediately apparent that not accounting for vertical velocities can introduce large errors in wind direction estimation at ridge height. Vertical velocities as large as $2\ ms^{-1}$ were often observed in the vertical stare, which can cause errors in wind direction estimation to be near 40 degrees if the retrievals only use two DLs. Due to the close proximity to CLAMPS, VT3 would have been the most erroneous 2D retrieval. However, it was possible to retrieve the 3D winds from this location.

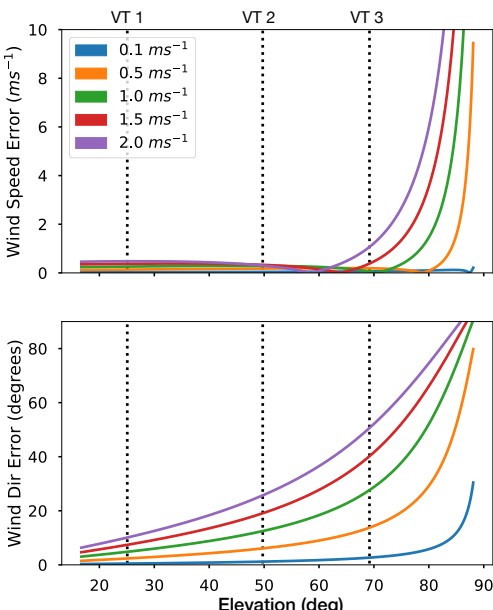

**Figure 4.** Wind speed and direction errors associated with neglecting the vertical velocity in 2D virtual towers. Results are for R=300m and H=200 as depicted in Figure 3. These values were chosen to mimic virtual tower measurements slightly above ridge height. The line colors correspond to different vertical velocities examined. Note that as vertical velocity increases, error in wind direction estimation increases even at relatively low elevation angles.



### 3.3 Selected Case Studies

To help illustrate the strengths and weakness of the retrieval methods, data from a few different cases selected from the Perdigão IOP were analyzed. The selected cases were subjectively identified based on the analysis of DL scans and stability profiles. Additionally, data availability was taken into account. For the following analysis, three cases were considered: one

with spatially homogeneous flow with little vertical velocity, one with spatially homogeneous flow with large vertical velocities, and one case with spatially heterogeneous flow and complex flow interactions.

By looking at cases that are more spatially homogeneous, it was possible to put more trust into the single-Doppler wind retrieval methods. This allowed more robust comparisons to insure the accuracy of the virtual tower retrievals. Finding cases with different amounts of vertical motion while still being quasi-homogeneous made it possible to analyze the amount of error

vertical motion introduced into the 2D retrievals. Once the virtual tower uncertainties were well characterized, it was then possible to find a highly complex case to illustrate how the virtual towers can be used to understand complex flow phenomena.

#### 3.3.1 Quasi-homogeneous Flow With Limited Vertical Motion

We selected a day with moderate to high wind speeds at ridge height to minimize the spatial variability of the flow. Additionally, we target a time period during which the observed vertical velocities were small by examining the vertical stare from the

CLAMPS DL. June 12, 2017 from 0-6 UTC fit these criteria well.

During this period, a short wave trough was off the coast of Portugal. Winds at 500 mb were approximately 20 kts from the SW. No discernible mesoscale surface features were present as a result of the trough. However, there are a number of mesoscale circulations that dominate the flow around Perdigão (Fernando et al., 2018). Thermal flows from the Serra de Estrela to the North often compete against synoptic scale flows. This can introduce unique layering of wind speeds and direction near the

ground. Winds during this period were 5-10 $ms^{-1}$ throughout the night (Figure 5b) and were out of the northeast (Figure 5e). A strong temperature inversion was present in the valley. Additionally, there was very little vertical motion and turbulence (Figure 5h and 5k). To insure the flow was spatially homogeneous despite the thermal flows and complex thermodynamics, RHI scans were all visually inspected.

#### 3.3.2 Quasi-homogeneous Flow With Strong Vertical Motion

Similar to the previous case, we again targeted a time period with moderate to high wind speeds during which the flow along the ridges can be assumed to be quasi-homogeneous. However, this time we selected a time period with strong vertical motions. This allowed instantaneous double- and triple-Doppler virtual tower measurements to be easily compared to single-Doppler measurements (such as VADs). Additionally, the error analysis from Section 3.2 could be validated using the 3D tower. During the overnight hours of June 14, 2017, the shortwave trough mentioned in Section 3.3.1 had moved over the area. The valley

was neutrally stratified through the night. Winds after 3 UTC increased to approximately 10 $ms^{-1}$ from the southwest (Figure 5c) and a persistent wave formed, causing large and steady vertical velocities over the CLAMPS site (Figure 5i and 5l). The higher winds mean it is more likely that the wind field was homogeneous along the length of the valley.




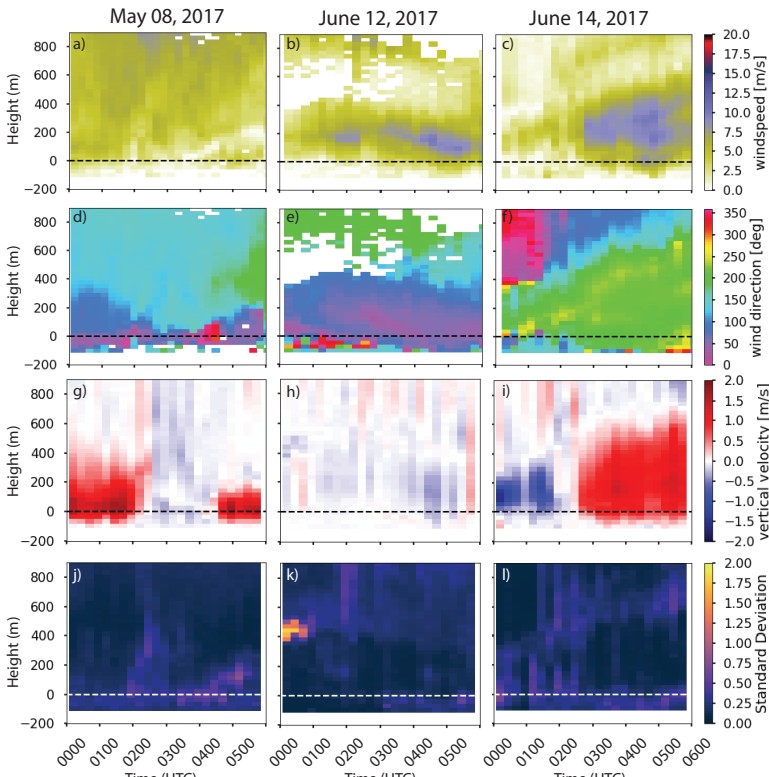

**Figure 5.** Time-height plots of three nights observed by the CLAMPS system. The top row (a-c) is wind speed derived from the 70 degree VAD scan, second row (d-f) are the wind directions from the VAD, the third row (g-i) are 15-minute averages of vertical velocity derived from the CLAMPS vertical stare data, and the bottom row (j-l) is the standard deviation from the averages. The left column shows data from May 8, 2017, the middle column data from June 12, 2017, and the right column for June 14, 2017. The height axis is relative to the base of the wind turbine on the southwest ridge, which is a proxy for ridge height.

### 3.3.3 Spatially Complex Case

While the two previous cases were selected with the intend of assessing the retrieval methods, the last case was chosen to illustrate how useful the virtual towers can be for measuring the spatial variability of features along the valley. A moderately complex case, May 8th 2017, was selected. During this time, there was a low pressure system off the coast of the Iberian Peninsula and a ridge over Perdigão. Aloft, winds over the IOP site were from the southwest at 30 kts around 500 mb.

The complexity of the flow for May 7-8th is shown in Figure 5. Generally, winds are from the northeast during the night of the 7th. On the 8th, winds veered through the night and became very weak at ridge height (Figure 5a and 5d). Wind directions





are quite variable since the mean wind is interacting with various slope flows occurring inside the valley. Numerous periods of strong vertical motion were observed over the CLAMPS site due to waves forming as a result of the terrain. On the 8th, a unique, double wave feature was observed along with a re-circulation within the valley. This occurred during the period of relatively low wind speeds around 05 UTC (Figure 5a). The double wave feature did not appear often during the rest of the IOP

and lends itself well as a good test for the virtual tower retrievals, especially the 3D virtual tower, given its complex evolution over time and space.

## 4   Results

### 4.1   Comparison of 2D and 3D Virtual Towers for Quasi-homogeneous cases

By looking closely at the cases presented in Sections 3.3.1 and 3.3.2, it is possible to determine whether or not the errors

hypothesized in Section 3.2 are noticeably present in the data. For this analysis, VT3 will be considered to be the truth since it has the fewest number of assumptions applied and contains the full 3D wind vector. Additionally, the CLAMPS VAD and CU DBS scans are expected to perform reasonably well in these conditions.

Starting with the most simple case, quasi-homogeneous flow with little vertical velocity, it is possible to directly compare the retrieval methods with minimal violations to the underlying assumptions. Unfortunately, VT4 and VT1 did not meet the

retrieval criteria because the CLMAPS DL became slightly out of sync with WS5 and WS6 and the time difference between scans was too large. However, for this application, having VT2 and VT3 will suffice since they are the closest to each other spatially. Data from these two virtual towers and CLAMPS VAD are shown in Figure 6. There is near perfect agreement between VT3 and VT2, which is to be expected since there is very little vertical velocity to introduce error into VT2. There are some slight differences from -100 to 0 m, but there were also slight vertical velocities measured in VT3 and in the vertical

stare. The areas with the least amount of vertical motion (e.g. around 120 m) have the greatest agreement in wind speed and direction. Figure 7b and e shows the time series of the wind speed and direction at the 100 m point on VT3 compared to the 100 m wind speed and direction (5 minute average) from TRSE09. Overall, there was really good agreement between the two systems, even at low wind speeds.

By contrast, looking at the June 14 case presented in Section 3.3.2 where higher vertical velocities were measured, consistent

differences between the 2D and 3D retrievals can be noted. Based on Figure 4, one would expect there to be approximately a 10- to 20-degree difference in wind direction between the fully resolved 3D tower (VT3) and the 2D tower (VT2) at 100 m above ridge height, where the vertical velocity was approximately 1 $ms^{-1}$. Figure 8 shows that the wind direction from VT2 differ by 20° from the wind direction at the CLAMPS VAD and VT3, which lends credit to the previously discussed idealized model and provides a simple way to estimate the uncertainty in the 2D towers due to the vertical motion. There are also slight

differences in wind speed present. While VT3 and the CLAMPS VAD agree well throughout the profile, VT2 is consistently indicating 1 $ms^{-1}$ higher wind speeds, which agrees well with the expected offset in Figure 4. Again, the time series of VT3 and VT2 agree qualitatively well with the tower data (Figure 7c and f).





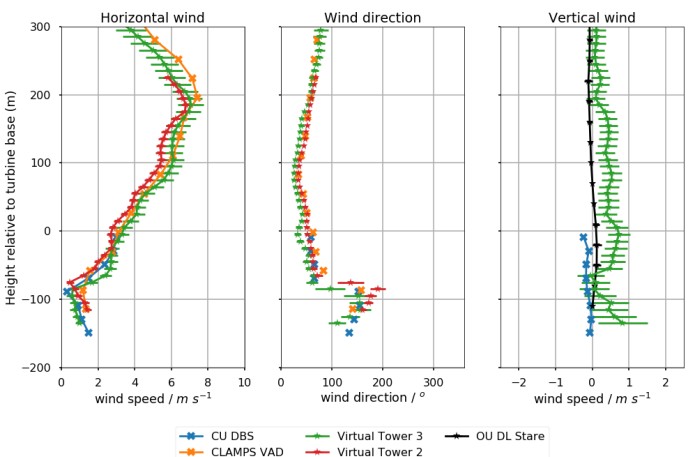

**Figure 6.** Profiles of the CU DBS (blue), CLAMPS VAD (orange), VT2 (red, 2D), VT3 (green, 3D), and the vertical stare from the CLAMPS DL (black) from June 12, 2017 at 2:55 UTC (discussed in Section 3.3.1). The error bars on the OU DL Stare profile are the standard deviation of the vertical velocities observed between 1 minute before and 1 minute after the virtual tower time. VT4 and VT1 are not available for this time period due to DL malfunction and not meeting retrieval criteria, respectively. Note the height axis is relative to the base of the wind turbine, which is a proxy for the height of the ridge.

## 4.2 Comparison of Virtual Towers and VAD/DBS Retrievals for Spatially Variable Flow

As mentioned previously, the traditional DL technique for retrieving the wind speed and direction with a single DL is the VAD or the DBS technique. However, in the overnight hours of the Perdigão campaign, the assumption of horizontal homogeneity is often violated due to flow phenomena created by the terrain. This is expected to be particularly prominent at the lowest levels
5 of the VAD or DBS profiles near the surface. We will now examine data from May 8 (Section 3.3.3) to assess the virtual towers in weak, highly variable flow. Using the spatially distributed virtual towers can be helpful to study these type of flows.

Figure 9 illustrates how single DL retrievals can break down in complex terrain. A few things become apparent from the comparison. In general, the CLAMPS VAD and the virtual towers agree well in wind speed and direction well above the ridge. Looking closer at retrievals above ridge height, there seems to be a consistent offset between VT3 and each of the other 2D
10 towers, which can be explained by the vertical velocities, as discussed in the previous section. This offset gets larger as the profile approaches ridge height where the vertical velocities are larger. However, well below top of the ridge, the physical towers indicate that vertical velocities are small, so the 2D towers can be trusted inside the valley during this time period.

Around 100 m below ridge height, the different techniques start to diverge. Wind speeds stay relatively consistent between each technique below ridge height, but the directions are highly variable. In particular, the wind directions from the CU DBS





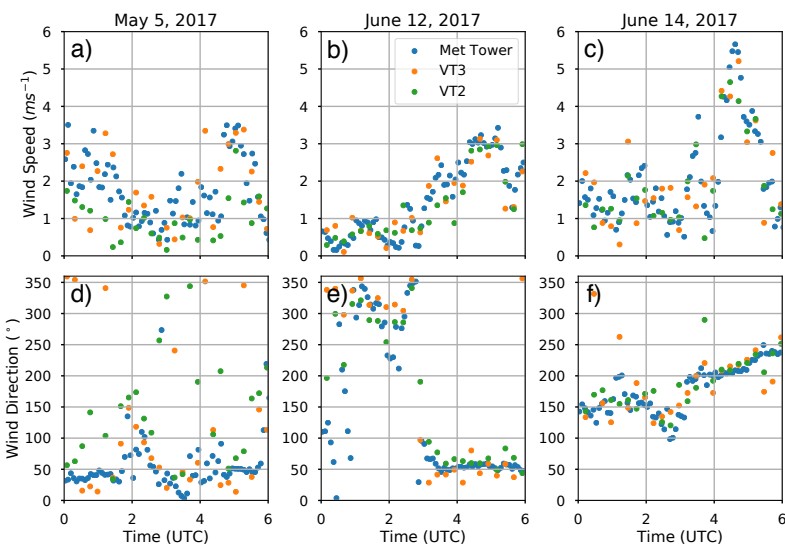

**Figure 7.** Time series of wind speeds (a-c) and wind directions (d-f) from VT3 (orange points), VT2 (green points), and the 5 minute averaged data from TRSE09 (blue points) at 100 m for the three selected cases.

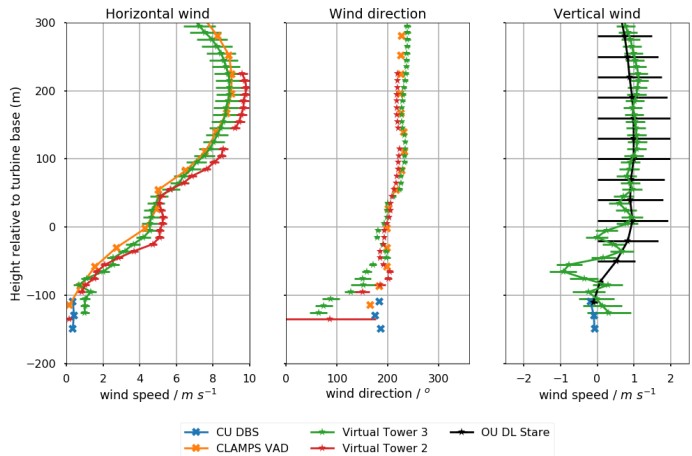

**Figure 8.** Same as Figure 6, but for June 14 at 2:58 UTC when higher vertical velocities were observed.





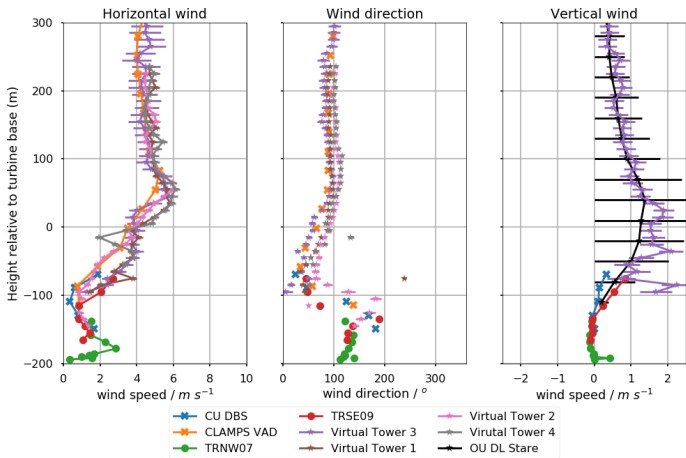

**Figure 9.** Similar to Figure 6, only with meteorological tower data added. Profiles are from May 8 at 00:32 UTC. Single Doppler retrievals are denoted with Xs, meteorological towers denoted by dots, virtual towers are denoted by asterisks. In general, the methods agree well above the ridge, but results begin to get noisy 100 m below ridge height.

scan differ greatly from the towers, both virtual and physical. The CLAMPS VAD tends to agree slightly more, but still differs significantly from the collocated VT3 just below ridge height.

The vertical velocities from VT3 and the OU DL vertical stare also agree quite well above ridge height. Around the same level as the wind speeds and directions, the stare and the 3D virtual tower start to diverge. The differences could be due to each measurement representing a different time. The tower data are 5 minute averages, while the OU DL stare data and the CLAMPS/DLR virtual tower are instantaneous measurements at slightly different locations, so some differences are to be expected.

Breakdowns in the single Doppler DL retrievals can be observed by closely examining Figures 6 and 8, assuming VT3 is taken to be the most accurate representation of the wind speed and direction. Similar to the highly complex case, there is relatively good agreement well above ridge height, where the flow is less affected by the terrain. Inside the valley however, the flow is much less horizontally homogeneous due to the complexity of the valley floor. This causes there to be large differences between the virtual towers and the single Doppler retrievals (for example, below -100 m in Figure 8).

In summary, Figure 9 shows how all the towers (both virtual and real) compare to one other. In general, the wind speed all line up nicely. There are differences in the wind direction though. The differences can be attributed to the error associated with the 3D towers discussed in Sections 3.2 and 4.1. As vertical velocities become larger at and slightly below ridge height, the spread in wind direction gets slightly larger. Additionally, the spread is approximately what would be expected from Figure 4. However, there is still noticeable spread in the lowest levels that is likely due to heterogeneity of the terrain. Additionally, Figure




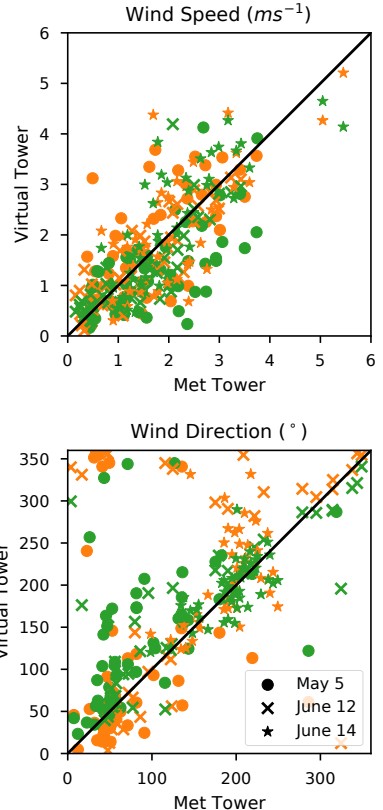

**Figure 10.** Scatter plots of the wind speeds (top) and direction(bottom) from VT2 (green) and VT3 (orange) vs the 5 minute averages of wind speed and direction from TRSE09.

7 shows that VT3 agrees well throughout the night, however VT2 contains a larger spread in wind directions. To determine if these are real, the flow needs to be examined in a spatial sense. Taking into consideration all the sources of uncertainty in the towers, it is possible to examine the flow in and around the valley at a high level of detail. Overall, the virtual towers perform reasonably well during each case study (Figure 10). A slight positive bias in wind direction was present in VT2 since it suffers
5 from the errors described in previous sections.

### 4.3 Spatial Analysis

One of the main reasons Vale do Cobrão was chosen for the experiment was the quasi-two-dimensional nature of the ridges; they were thought to be the best way to represents a series of periodic rolling hills and that flow perpendicular to the ridges



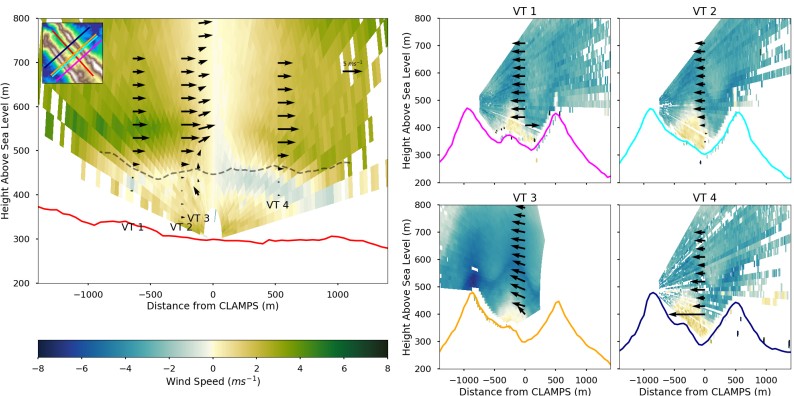

**Figure 11.** Left: Along-valley cross section of the flow for May 8th at 00:32 UTC. The color fill is the radial velocity projected into the horizontal from the OU RHI used to create the virtual towers. Positive values indicate flow in the +X direction, which is directed to the NW. The overlaid vectors are the components of the virtual towers projected into the plane of the RHI scan, the inset plot shows how the terrain cross sections are oriented and where they are located, and the grey dashed line is the height of the SW ridge. Right: Similar to the along-valley plot on the left, only in the cross valley sense. +X here points to the NE. The color of the line matches the cross section indicated in the terrain inset. This plot corresponds to the profiles shown in Figure 9.

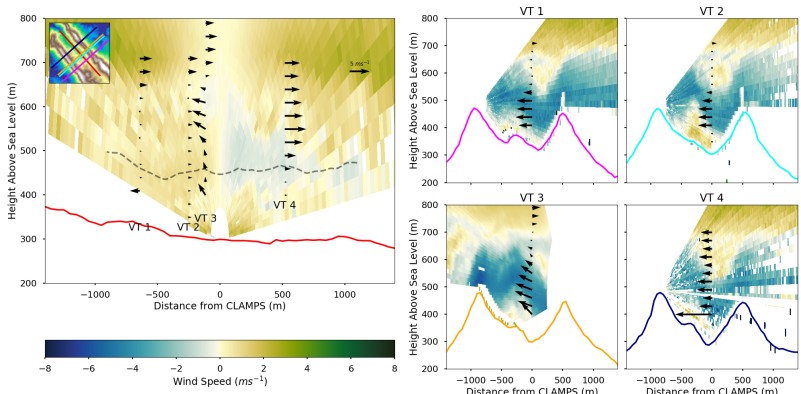

**Figure 12.** Same as Figure 11, but for 5:03 UTC. Note the different structure of the wave and the shear layers in the cross valley winds. Also note the different strengths of the recirculations.

could also be considered quasi-2D. While this during some nights, periods with lower wind speeds tend to have more spatial heterogeneity. In order to visualize this heterogeneity, it is best to think of wind components in the cross-valley and along-valley sense.



Analysis of the virtual towers coupled with the cross-valley RHIs from the DLR and DTU DLs often shows that the flow can not be considered entirely 2D, especially at lower wind speeds. In Figure 11, cross valley flow is approximately -4 $ms^{-1}$. No clear jet is present in the RHI or virtual towers, but there is a large recirculation in the valley. During this time period, flow was strong enough that the circulation was present in all the cross sections of the valley. However, the size and strength of the recirculation varied due to the topography of the valley floor. Where the recirculation is strongest, cross valley wind speeds are slightly reduced in the virtual tower (VT2 and VT4). This is especially prominent in the lowest levels of the virtual towers near the circulation. This also appears in some of the levels well above the valley, which is significant because it implies that the recirculation is correlated with changes in the flow well away from the feature itself.

Looking at the along-valley component of the flow, winds are very calm and near zero within the valley. There does appear to be a weak jet just above the ridge at 550 m. There is evidence of a wind speed max near the surface, which indicates there is a downslope flow. This feature was often observed overnight in the along valley RHI from CLAMPS. Though the virtual towers do not extend far enough into the valley to capture this flow, the meteorological towers are able to capture the near surface speed max associated with it (Figure 9). This persistent flow adds to the three-dimensionality of the flows present in the valley.

Analysis of a different time period during the night of May 8th further shows the complexity of the flows in the terrain. During this night, winds veered rather quickly aloft and weakened closer to the surface (Figure 5). An interesting transition occurs and a unique double wave forms within the valley along with a shear layer (Figure 12). Like the previous case, there is little along-valley component of the wind below the ridge. There is still a speed max in the down-valley flow in the lowest 20-30 m of the OU RHI scan, which is indicative of the aforementioned downslope flow. The recirculation is also still present inside the valley and has a highly complex structure that differs based on the cross section of the valley. The recirculation looks to be largest in the VT2 cross section, similar to the previous case. This again appears to slow wind speeds upstream of the recirculation.

## 5  Conclusions

Multi-Doppler analyses are a useful tool for understanding and quantifying wind characteristics in complex terrain. Though the scans used for the virtual towers were uncoordinated, they are useful for diagnosing flow conditions in and above the valley. The virtual towers help fill the gap in wind speed measurements inside the valley above the height of the physical towers (100 m) and where more traditional DL scanning strategies may not be fully valid given the complexity of the site and in particular for this experiment provide nicely distributed measurements along the valley at heights which are not captured by any other instrument.

Though the virtual towers are well suited to study the complex flows observed during the IOP, they are not without limitations. The uncertainty in the radial velocities needed to be propagated through the retrieval. Due to the positioning of the DLs used for the virtual towers, this meant that uncertainty in the horizontal wind retrieval was larger with increased height. However, vertical velocity retrievals on the single 3D virtual tower became more certain with height as a larger component





of the vertical velocity was observed in the radial velocities at the higher elevation angles. Additionally, the 2D virtual towers made the assumption that there was no vertical velocity, which is often violated in this terrain. Due to this, they are prone to errors, particularly in wind direction. It was shown that these errors can be estimated and accounted for in an analysis.

In terms of wind direction, despite the uncertainties in the 2D retrievals, the virtual towers agree better with the meteoro-
logical towers situated inside the valley than the VAD/DBS scans at those levels. However in the analyzed cases wind speeds at these levels were quite small, so a more detailed inter-comparison between all the different methods of wind estimation is needed. This could be done with a DL simulator that is able to use fine-scale model output from the valley to mimic DL radial velocities and noise. These could then be fed into the virtual tower retrieval and compared directly to the model output.

Combining virtual tower data with meteorological towers on the ridge allows wind speeds to be sampled as cross-valley flow
enters, goes through, and exits the valley and give a more complete picture of the spatial evolution of features in complex terrain. They also allow a more comprehensive validation dataset for numerical models in the future. While simulated RHI scans from numerical models can be compared to the observed RHIs, only the combination of multiple DLs allows the retrieval of the 2D and 3D wind vector. These data could one day be fed into a model-based wind retrieval using advanced data assimilation methods to estimate the full 3D wind field in and around the valley to gain more insight into the governing physics of the flow
to gain more insight into the governing physics of the flow.

*Data availability.* Data of all instruments that were used in this study is stored on three mirrored servers owned by DTU, University of Porto and the NCAR Earth Observing Laboratory (EOL) respectively. The data is publicly available through dedicated web portals of the University of Porto (https://windsp.fe.up.pt/) and EOL (http://data.eol.ucar.edu/master_list/?project=PERDIGAO).

*Acknowledgements.* We would like to thank José Palma and José Carlos Matos for their work to make this experiment a success. We would
also like to thank Matt Carney and Edward Creegan for their hard work in getting CLAMPS running in Perdigão despite many initial challenges and we thank the entire Perdigão team for their collaborative spirit before, during, and after the campaign. Lastly, we appreciate the hospitality of the people in the municipality of Alvaiade. This research was funded by NSF grant AGS 1565539 and by the Federal Ministry of Economy and Energy on the basis of a resolution of the German Bundestag under the contract numbers 0325518 and 0325936A.



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
