# Peer review of "Analysis of Flow in Complex Terrain Using Multi-Doppler Lidar Retrievals"

_Atmospheric Measurement Techniques, 2018_

## Referee Comment (RC1) · Anonymous Referee #1 · 30 Apr 2019

The study tries to answer the question "how well the 2D and 3D multi-Doppler measurements perform in a complex terrain?" but does not give quantitative results in detail (or not fully comment on the results). There are assumptions made without full explanation which makes the methodology unrepeatable.

I believe the paper needs major changes. I would be happy to read again after the corrections.

- Section 2: If you want your paper to be easy to understand for everyone, even for people who does not know Perdigao experiment, I think you should make more figures showing the scanning patterns for the lidars. I think section 2 needs more then Figure 1.

[Figure]

- Page 2 Line 11: The term "large spatial heterogeneity" is not quantitative. Since the referred studies gives a range of number, I think you should rephrase.

- Page 6 Line 1: How do you justify using "linear interpolation"? Add references and comment on the contribution of the linear interpolation on the general uncertainty

- Page 6 Line 20: Even expert opinions need citation. You must either write more about your assumption of 2m/s and its reasoning in detail, or you need to cite a study doing so. It is a very critical step in your method chain, and you are making the method unrepeatable by just giving an assumption for your specific case. You must clear this point.

- Page 9 Line 16 + Page 10 Line 5: Journal uses SI units.

- Page 13 Figure 8: I cannot understand the difference of the vertical wind speed values above 200m. OU DL and VT3 show a big difference. Why? Can you comment?

- Conclusion: I think more discussion is required for the limitations of the lidars.

---

## Referee Comment (RC2) · Anonymous Referee #2 · 1 Jul 2019

The manuscript presents interesting analysis approaches on the topic of spatial variability wind effects, which is an important one. The text is often hard to follow, though, and I had to reread the paper a couple of times to understand even the main thrusts. I believe the manuscript needs a major rewriting, especially in the following areas: a significant rearrangement of the text, elimination of vague and qualitative statements, and addressing what seem to me to be places where the text does not match the analyses presented. In several places, the last point may be addressable (or at least clarified) by plotting difference profiles in addition to the mean profile, and adjusting the color scales so the reader can see what the authors are seeing.

Major recommendations: 1. Text rearrangement: This is not a long paper, so the brief descriptions of the three types of flow case studies in Section 3.3, followed by the

case studies themselves in Sections 4.1 and 4.2, seemed like jumping around and was confusing. The descriptions in 3.3 sounded more like introductions to sections than 'Methods.' This was made a bit more confusing because the cases are presented in a different order from the columns of Fig. 5, which is an introductory and summary figure. I recommend combining subsection 3.3 and Section 4â ĂŤby making 3.3 into the beginning of Section 4, with the first 3 subsections being the three case studies. The first subsection (4.1, 12 June) would start with the text from 3.3.1 (lines 13-24 from p.9) and then add the text from p.11, lines 13-23; 4.2 (14 June) would start with text from 3.3.2 and add that from p.11, lines 24-32; and similarly for 8 May into a new 4.3. If Fig. 5 were rearranged to show 8 May in the last column, the text would go smoothly into the material in Section 4.3, either as its own section (new 4.4) or added to the 8 May material in the new 4.3.

2. Qualitative descriptions. Many of the descriptions are vague and qualitative, and need to be made quantitative. Some examplesâ ĂŤ'moderate to high wind speeds' (p.9, lines 13 and 25), 'became very weak' (p.10, line 7); 'very little vertical velocity;' 'slight differences,' 'agree qualitatively well with. . .' (p.11, lines 13, 30, 32); 'relatively good agreement well above ridgetop' [what levels?], and 'large differences,' and 'wind speed [sic.] all line up nicely' (p.14, lines 10, 11, and 13), etc. Small differences, strong or weak winds, all depend on application, so it is important to specify what these statements mean quantitatively.

3. Mismatch between text and analyses. In some places, the description in the text does not seem to me to be supported by the evidence presented in the analysis figures. Examples: - (p.11, lines 12 and 20) 'near perfect agreement between VT3 and VT2. . .' apparently referring to the red and green curves in Fig. 6, 'near perfect' seems like a big exaggeration, and these two curves seem to be off by $\sim \frac{1}{2}$ m/s through most of their depth. 'the areas with the least amount of vertical motion (e.g., around 120 m) have the greatest agreement. . .' There seems to be about $\frac{1}{2}$ m/s difference at 120m, which looks to be about as big as this difference gets. Smaller difference values seem to be

at -30 m and 180 m, where vertical velocities do not seem smallest, although it's hard to tell with the wind-speed scales shown. (p.14, lines 13 and 15) 'wind speed all line up nicely,' and 'spread in wind direction gets slightly larger.' These are very hard to see, and so it's hard to know whether they are significant enough to be concerned about. Some of these may just be difficult to see because of the scales plotted, and it's asking a lot of a reader to be able to see differences that may be not much larger than the widths of the lines. Such as (p.12, line 9): 'a consistent offset between VT3 and each of the other. . .' is hard to see. So I suggest plotting vertical profiles of the differences, where they can be seen on an expanded scale. Similarly, please compress the color scale in the top row of Fig. 5 (like, max values of 12 or 15 m/s instead of 20) to show more of the structure described in the text. Overall, please check carefully that all the claims in the text are supported by the Figures used as evidence.

4. Homogeneous. Homogeneous means invariant in space ('spatially inhomogeneous' is redundant), including the vertical, so the best that can be hoped for in the atmosphere is horizontal homogeneity. The flow within complex terrain such as Perdigão is very seldom horizontally homogeneous, especially at night. So, claims that either June case has horizontally homogeneous flow must have much more justification, or the wording needs to be changed (like, less inhomogeneous, or, less horizontal variability, etc.).

Minor comments: 5. The appropriate manner of scanning for and computing mean wind profiles depends on application—there is not a 'one-size-fits-all' solution, and each manner (single lidar, multiple lidar, etc.) has advantages and drawbacks. As this is a relatively new field, I have not heard of robust studies that convincingly and quantitatively demonstrate that one approach is clearly better than others for all applications and circumstances. Authors seem to be favoring the virtual tower approach in their review, but I would encourage them to take a more neutral tone in their review, along the lines of 'there are several approaches, and each has advantages and disadvantages' as just stated. VAD has the possibility of inhomogeneities across the scan adding to

the uncertainty of the calculation, but according a recent Mann et al. paper, virtual towers can have difficulties in the scanning/ data acquisition stage with beam alignment and coordination, leading to uncertainties especially in complex terrain. 6. Throughout, there are a number of places where is seems to be assumed that the only way the speeds, and especially directions, of different scanning approaches can be different is because of vertical velocity. In a complex terrain setting, there can always be many different terrain-related and other reasons. The authors should list these reasons, and when discussing differences, say something like that the differences are consistent with a possible vertical-velocity explanation. 7. Add a 'Site Elevation ASL' column to Table 1. 8. Add a Table 2 of virtual towers, specifying which lidars are participating in each tower. It's sometimes hard for a reader to keep track of. 9. (p.3, line 10: 'To validate numerical models, detailed measurements of the flow at multiple scales are required.' I suggest references here, since in my experience many modelers are not necessarily in agreement about this. Banta et al. 2013 and Fernando et al. 2015, both BAMS, are a couple of papers that make this point, I think. 10. p.4, line 15 and p.5, line 6: 'lower orange site' what is this? No context for this. 11. p.5, about line 1: what is the minimum height (minimum range) of the CLAMPS lidar in vertical staring model? 12. p.7, Fig. 3: can hardly read the labels. Also true of many other figs—please make sure labels are big enough to be legible. 'OU DL' in caption – is this CLAMPS? Please stick to one name for each lidar—there are a lot of lidars to keep track of. 13. Section 3.2: The explanation of what lidar is doing what, and how that impacts the horizontal wind uncertainty is not clear to me. What do the vertical lines marked VT1, VT2, and VT3 mean – please clarify in caption. 14. Sections 3.3.3 and 4.3: Figs. 11, 12 are almost certainly hydraulic jumplike mountain lee-wave features, which at larger scales would be downslope windstorms. An early paper with lidar cross sections is Clark et al. 1994 JAS. 15. A couple of places had kts for wind speed, instead of m/s.

---

## Author Comment (AC1) · 27 Aug 2019

We thank Anonymous Reviewer #1 for providing their review on our manuscript. The attached .zip archive contains our response and, in a separate file, a latexdiff version of our revised manuscript.

Please also note the supplement to this comment: https://www.atmos-meas-tech-discuss.net/amt-2018-417/amt-2018-417-AC1-supplement.zip

---

## Author Response (AR1)

*We thank Anonymous Reviewer #1and #2 for providing their review on our manuscript. Our responses to each reviewer are below. Additionally, a latexdiff of the new manuscript is attached to this document.*

Reviewer 1

*The authors thank the reviewer for their comments on the paper. A common theme from both reviews was the overuse of qualitative descriptions and analysis. Care has been taken to eliminate these from the text. Below you will find responses to each individual comment.*

**The study tries to answer the question "how well the 2D and 3D multi-Doppler measurements perform in a complex terrain?" but does not give quantitative results in detail (or not fully comment on the results). There are assumptions made without full explanation which makes the methodology unrepeatable. I believe the paper needs major changes. I would be happy to read again after the corrections.**

*We thank the reviewer the feedback. We have expanded the description of the methodology that we used and have made additional major changes to the manuscript which are described in detail thereafter.*

**- Section 2: If you want your paper to be easy to understand for everyone, even for people who does not know Perdigao experiment, I think you should make more figures showing the scanning patterns for the lidars. I think section 2 needs more then Figure 1**

*The authors agree that fully grasping the instrument layout can be challenging. However, it is this complexity that makes it difficult to convey the scanning strategy of each individual Doppler lidar (DL). To address the reviewer's concern, DLs that were not used in this study have been removed from Figure 1. Additionally, we added a column to Table 1 that indicates which DL's are used in each of the virtual tower (VT) retrievals. This information, together with the exact values of the azimuth and elevations of the range-height indicator (RHI) scans for each DL provide detailed information about the scanning patterns and the instrument layout. Furthermore, we have added references to the Perdigão overview paper (Fernando et al., 2019) which provides a complete overview of the Perdigão experiment and payed attention to using site names that are consistent with the descriptions in this paper.*

**- Page 2 Line 11: The term "large spatial heterogeneity" is not quantitative. Since the referred studies gives a range of number, I think you should rephrase.**

*This has been reworded to better address the issues that arise when using DL scans in complex terrain and to include quantitative statements that are guided by the cited literature. We have also made an effort to remove qualitative descriptions throughout the rest of the text.*

**- Page 6 Line 1: How do you justify using "linear interpolation"? Add references and comment on the contribution of the linear interpolation on the general uncertainty**

*The range gates of the lidars at the location of the virtual towers are comparatively dense (Figure 1). The largest distance of an interpolation point to an actual range gate is 16 m for the CLAMPS lidar, 5 m for the two DLR lidars and smaller than 10 m for the DTU lidars:*

- *CLAMPS, VT1: 15.9 m*
- *CLAMPS, VT2: 16.3 m*
- *CLAMPS, VT3: 16.2 m*
- *CLAMPS, VT4: 13.7 m*
- *DLR#1: 5.2 m*
- *DLR#2: 4.8 m*
- *WS2: 8.9 m*
- *WS5: 10.0 m*
- *WS6: 9.5 m*

*We have evaluated the difference of cubic spline interpolation to linear interpolation with example profiles (Figure R2) and found that differences of less than 0.1 m s^{-1} occur (Figure R3), which is well within the uncertainty we set for the lidar radial wind speed measurement. Since neither cubic splines nor linearity can be assumed for the atmospheric flow field, we decided for the simpler, computationally more efficient method of linear interpolation.*

[Figure]

*Figure R1: Visualization of lidar range gate centers of the CLAMPS RHI (blue dots) and the interpolated points for the virtual tower retrieval at location VT3 (black squares).*

[Figure]

*Figure R2: Comparison of interpolation of CLAMPS radial wind speed measurements to the VT locations using a linear interpolation (blue line and dots) and a cubic spline interpolation (orange line and dots)*

[Figure]

*Figure R3: Difference between linear interpolation and cubic spline interpolation of the CLAMPS radial wind speeds at the four VT locations.*

**- Page 6 Line 20: Even expert opinions need citation. You must either write more about your assumption of 2m/s and its reasoning in detail, or you need to cite a study doing so. It is a very critical step in your method chain, and you are making the method unrepeatable by just giving an assumption for your specific case. You must clear this point.**

*We would like to point out that the assumed accuracy was 0.2 m/s not 2 m/s, we apologize for any confusion may have caused (the original paper stated that we used .2 m/s). This has been addressed in the text by adding citations to studies that have analyzed this problem. Though the specifications of the Windcube 200s has a specified accuracy of 0.5 m/s, Kigle (2017) found they were accurate to ~0.10-0.15 m/s in comparison to a sonic anemometer at a meteorological mast during the Perdigão campaign. Therefore, a conservative value of 0.2 m/s was used for this analysis.*

**- Page 9 Line 16 + Page 10 Line 5: Journal uses SI units.**

*Fixed*

**- Page 13 Figure 8: I cannot understand the difference of the vertical wind speed values above 200m. OU DL and VT3 show a big difference. Why? Can you comment?**

*The authors are unsure what differences are being highlighted by this comment. If the reviewer is referencing the differences in vertical velocity just below ridge height in Figure 8, this could be due to the virtual tower being considered an instantaneous measurement, while both the OU DL Stare and the physical towers are averages. Depending on how steady the flow is, there could be some differences between the instantaneous measurements and the averaged measurements (as noted in p15, lines 2-4)*

**- Conclusion: I think more discussion is required for the limitations of the lidars.**

*We mention some of the downfalls of multi-Doppler retrieval methods toward the end of the original introduction and we have slightly revised this to favor virtual towers less. Additionally, we already discussed limitations of the use of uncoordinated RHI scans in our virtual tower retrievals in the conclusions. This discussion has been further expanded to better highlight the limitations of the various profiling techniques (p18, lines 18-21)*

Reviewer 2

*The authors thank the reviewer for their comments on the paper. A common theme from both reviews was the overuse of qualitative descriptions and analysis. Care has been taken to eliminate these from the text. Below you will find responses to each individual comment.*

**The manuscript presents interesting analysis approaches on the topic of spatial variability wind effects, which is an important one. The text is often hard to follow, though, and I had to reread the paper a couple of times to understand even the main thrusts. I believe the manuscript needs a major rewriting, especially in the following areas: a significant rearrangement of the text, elimination of vague and qualitative statements, and addressing what seem to me to be places where the text does not match the analyses presented. In several places, the last point may be addressable (or at least clarified) by plotting difference profiles in addition to the mean profile, and adjusting the color scales so the reader can see what the authors are seeing.**

*We thank the reviewer for the constructive feedback and agree that the paper lacked clarity. We had initially not included difference plots because it is not necessarily obvious which data set should be chosen as reference (or in other words it is not obvious what the "truth" is) as none of the profile measurements are free of errors. We agree however, that adding difference plots provides a better picture of the values of the observed differences and we added such plots in the revised version.*

**Major recommendations:**

**1. Text rearrangement:**
- **This is not a long paper, so the brief descriptions of the three types of flow case studies in Section 3.3, followed by the case studies themselves in Sections 4.1 and 4.2, seemed like jumping around and was confusing. The descriptions in 3.3 sounded more like introductions to sections than 'Methods.'**
- **This was made a bit more confusing because the cases are presented in a different order from the columns of Fig. 5, which is an introductory and summary figure.**
- **I recommend combining subsection 3.3 and Section 4âA˘Tby making 3.3 into ˇ the beginning of Section 4, with the first 3 subsections being the three case studies. The first subsection (4.1, 12 June) would start with the text from 3.3.1 (lines 13-24 from p.9) and then add the text from p.11, lines 13-23; 4.2 (14 June) would start with text from 3.3.2 and add that from p.11, lines 24-32; and similarly for 8 May into a new 4.3.**
- **If Fig. 5 were rearranged to show 8 May in the last column, the text would go smoothly into the material in Section 4.3, either as its own section (new 4.4) or added to the 8 May material in the new 4.3.**

*The authors have made the change to the various sections as suggested to improve the readability of the text. Section 3.3 in the original text has been incorporated into section 4.1 and its subsections. Specific changes are as follows:*

1. *The introduction to the original 3.3 has been incorporated into the introduction of Section 4.1*
2. *Section 3.3.1 from the original text has been incorporated into Section 4.1.1*
3. *Section 3.3.2 from the original text has been incorporated into Section 4.1.2*
4. *Section 3.3.3 from the original text has been incorporated into Section 4.1.3*

*Additionally, Figures 5 and 7's columns have been rearranged to follow the sequence of cases that is used in the text, as opposed to chronological order.*

**2. Qualitative descriptions. Many of the descriptions are vague and qualitative and need to be made quantitative. Small differences, strong or weak winds, all depend on application, so it is important to specify what these statements mean quantitatively**

*Care has been taken to eliminate these examples as well as other examples where qualitative descriptions were used without context. Specific examples pointed out by the reviewer were changed as follows:*

- **moderate to high wind speeds' ˇ (p.9, lines 13 and 25),**
  - *wind speeds that exceeded 7-10 $ms^{-1}$ (p9, Line 2)(p11, line 6)*
- **'became very weak' (p.10, line 7);**
  - *were less than 3 $ms^{-1}$ at ridge height (p13, line 12)*
- **'very little vertical velocity;'**
  - *vertical velocities were largely less than 0.5 $ms^{-1}$ (p9, line 4)*
- **'slight differences,'**
  - *This has been either eliminated or clarified (e.g. p12, lines 10-12).*
- **'agree qualitatively well with. . .' (p.11, lines 13, 30, 32);**
  - *This has been clarified as being within the degrees of uncertainty (e.g. p12, lines 13-14)*
- **'relatively good agreement well above ridgetop' [what levels?],**
  - *greater than 100 m above the ridge (p14, lines 5-6)*
- **and 'large differences,'**
  - *This has been eliminated*
- **and 'wind speed [sic.] all line up nicely' (p.14, lines 10, 11, and 13), etc.**
  - *This has been clarified as being within the degrees of uncertainty*

**3. Mismatch between text and analyses. In some places, the description in the text does not seem to me to be supported by the evidence presented in the analysis figures. Examples:**
- **(p.11, lines 12 and 20) 'near perfect agreement between VT3 and VT2. . .' apparently referring to the red and green curves in Fig. 6, 'near perfect' seems like a big exaggeration, and these two curves seem to be off by ~ 1 2 m/s through most of their depth. 'the areas with the least amount of vertical motion (e.g., around 120 m) have the greatest agreement. . .' There seems to be about 1 2 m/s difference at 120m, which looks to be about as big as this difference gets. Smaller difference**

values seem to be at -30 m and 180 m, where vertical velocities do not seem smallest, although it's hard to tell with the wind-speed scales shown.

- **(p.14, lines 13 and 15) 'wind speed all line up nicely,' and 'spread in wind direction gets slightly larger.' These are very hard to see, and so it's hard to know whether they are significant enough to be concerned about. Some of these may just be difficult to see because of the scales plotted, and it's asking a lot of a reader to be able to see differences that may be not much larger than the widths of the lines. Such as (p.12, line 9): 'a consistent offset between VT3 and each of the other. . .' is hard to see. So I suggest plotting vertical profiles of the differences, where they can be seen on an expanded scale. Similarly, please compress the color scale in the top row of Fig. 5 (like, max values of 12 or 15 m/s instead of 20) to show more of the structure described in the text.**
- **Overall, please check carefully that all the claims in the text are supported by the Figures used as evidence.**

*These points have been addressed. In many of the given examples, it was difficult to see how well the different profiles compared. Therefore, Figures 6, 7, and 8 have been redone to include vertical profiles of the difference compared to VT3, as suggested by the reviewer. More care has been taken to describe these plots in a quantitative sense to make the results easier to understand. Figure 5's color scale has also been adjusted to better show features of the flow.*

**4. Homogeneous. Homogeneous means invariant in space ('spatially inhomogeneous' is redundant), including the vertical, so the best that can be hoped for in the atmosphere is horizontal homogeneity. The flow within complex terrain such as Perdigão is very seldom horizontally homogeneous, especially at night. So, claims that either June case has horizontally homogeneous flow must have much more justification, or the wording needs to be changed (like, less inhomogeneous, or, less horizontal variability, etc.).**

*This has been addressed throughout the text. The authors have made wording more specific. More care has been taken to describe the limits of the homogeneity when it is discussed. For example, in the June cases, the flow was largely horizontally homogeneous along the length of the ridges/valley due to the winds being perpendicular to the ridge/valley axis.*

**Minor comments:**

**5. The appropriate manner of scanning for and computing mean wind profiles depends on application; there is not a 'one-size-fits-all' solution, and each manner (single lidar, multiple lidar, etc.) has advantages and drawbacks. As this is a relatively new field, I have not heard of robust studies that convincingly and quantitatively demonstrate that one approach is clearly better than others for all applications and circumstances. Authors seem to be favoring the virtual tower approach in their review, but I would encourage them to take a more neutral tone in their review, along the lines of 'there are several approaches, and each has advantages and disadvantages' as just stated. VAD has the possibility of inhomogeneities across the scan adding to the uncertainty of the calculation, but according a recent Mann et al. paper, virtual towers can have difficulties in the scanning/ data**

**acquisition stage with beam alignment and coordination, leading to uncertainties especially in complex terrain.**

*The authors had mentioned some of the downfalls of multi-Doppler retrieval methods toward the end of the original introduction. Some key wording was changed/added to make it more apparent that multi-Doppler retrievals are not a guaranteed solution to measuring complex flows. In any case and for any method it is important to estimate the uncertainties of the retrievals in order to evaluate the results of comparative flow measurements as we have done in this study.*

**6. Throughout, there are a number of places where is seems to be assumed that the only way the speeds, and especially directions, of different scanning approaches can be different is because of vertical velocity. In a complex terrain setting, there can always be many different terrain-related and other reasons. The authors should list these reasons, and when discussing differences, say something like that the differences are consistent with a possible vertical-velocity explanation.**

*For single DL retrievals like the VAD and DBS scans, horizontal heterogeneity due to the terrain can certainly introduce errors into the retrievals along with the presence of a vertical velocity. In the case of the 2D VTs, the horizontal homogeneity assumption is not required since each DL is observing a single point in space. This difference has been clarified in the text. Additionally, discussion about flow inhomogeneities has been added on p18 lines 21-24.*

**7. Add a 'Site Elevation ASL' column to Table 1.**

*This has been included.*

**8. Add a Table 2 of virtual towers, specifying which lidars are participating in each tower. It's sometimes hard for a reader to keep track of.**

*Another column has been added to Table 1 that indicates which VT(s) each DL is used in.*

**9. (p.3, line 10: 'To validate numerical models, detailed measurements of the flow at multiple scales are required.' I suggest references here, since in my experience many modelers are not necessarily in agreement about this. Banta et al. 2013 and Fernando et al. 2015, both BAMS, are a couple of papers that make this point, I think.**

*These references have been added*

**10. p.4, line 15 and p.5, line 6: 'lower orange site' what is this? No context for this.**

*The wording of our site description has been updated to the "Orange Grove site" in order to match the Fernando et al. (2019) BAMs paper Perdigão overview paper.*

**11. p.5, about line 1: what is the minimum height (minimum range) of the CLAMPS lidar in vertical staring model?**

*This has been added to the text. With the configuration during Perdigão, the minimum range of the CLAMPS DL was ~75 m*

**12. p.7, Fig. 3: can hardly read the labels. Also true of many other figs. Please make sure labels ˇ are big enough to be legible. 'OU DL' in caption – is this CLAMPS? Please stick to one name for each lidar, there are a lot of lidars to keep track of.**

*The labels on Figure 3 and Figure 2 have been made larger. Additionally, care has been taken to make sure that labels and colors are consistent in text and plots for the various DLs/VTs/etc.*

**13. Section 3.2: The explanation of what lidar is doing what, and how that impacts the horizontal wind uncertainty is not clear to me. What do the vertical lines marked VT1, VT2, and VT3 mean – please clarify in caption.**

*This has been further clarified in the text and caption. The vertical lines are the elevations of the VTs for a given R and H (as shown in Figure 3), thus shows the expected error for that point given a vertical velocity.*

**14. Sections 3.3.3 and 4.3: Figs. 11, 12 are almost certainly hydraulic jumplike mountain lee-wave features, which at larger scales would be downslope windstorms. An early paper with lidar cross sections is Clark et al. 1994 JAS.**

*These certainly could be hydraulic jump type features. Though the authors could not locate the cited paper by the reviewer, Neiman et al. 1988 observed a flow reversal with Doppler lidar data. This reference and mention of it being a possible hydraulic jump has been added to p17 lines 8-9*

**15. A couple of places had kts for wind speed, instead of m/s.**

*This was fixed*

*Changes not in response to reviewer comments:*
  * *Figures 6, 8, and 9 in the original manuscript had an error with the error bars on the OU DL Stare profile. Instead of the standard deviation being plotted as the magnitude of the error bar, the mean was accidentally used as the magnitude. This has been fixed in the updated manuscript.*

[revised manuscript text omitted]

---

## Author Response (AR2)

Dear Jose,

Thank you for your feedback. We respectfully disagree with your assessment that major revisions were not performed on the manuscript. At the suggestion of the reviewers, our revisions had included a major reorganization of the text and significant additions to the analysis by adding new plots that show the differences between the virtual towers in Figures 6, 7, and 8. Additionally, care had been taken to eliminate vague and qualitative statements.

However, to address your concerns we took additional care to further improve the manuscript and implemented another round of major revisions (please see the latexdiff below for details). While some qualitative statements may remain, they are due to the lack of a true reference measurement. Such reference data would be necessary to fully validate the VTs qualitatively. This is now explicitly discussed in the new version of the manuscript. Additionally, to avoid any further confusion, the term "homogeneous" has been replaced with "Quasi-2D" when referencing some of the presented case studies, and we describe in detail the criteria applied when selecting these Quasi-2D cases.

Other minor changes include making unit formatting consistent, naming the meteorological towers consistently, and adding the DOIs for each dataset.

We hope you will find the new version to be much improved and meeting the standards for publication in AMT, or at least for sharing it one more time with the reviewers.

Best regards,

Tyler, Petra, Norman, Robert

[revised manuscript text omitted]